# Using representation balancing to learn conditional-average dose responses from clustered data

**Christopher Bockel-Rickermann**          *christopher.rickermann@kuleuven.be*
*KU Leuven*

**Toon Vanderschueren**          *toon.vanderschueren@kuleuven.be*
*KU Leuven*
*University of Antwerp*

**Jeroen Berrevoets**          *jeroen.berrevoets@maths.cam.ac.uk*
*University of Cambridge*

**Tim Verdonck**          *tim.verdonck@uantwerpen.be*
*University of Antwerp - imec*
*KU Leuven*

**Wouter Verbeke**          *wouter.verbeke@kuleuven.be*
*KU Leuven*

**Reviewed on OpenReview:** *https://openreview.net/forum?id=U8EMkndyq4*

## Abstract

Estimating the response to an intervention with an associated dose conditional on a unit's covariates, the "conditional-average dose response" (CADR), is a relevant task in a variety of domains, from healthcare to business, economics, and beyond. Estimating such a response is challenging for several reasons: Firstly, it typically needs to be estimated from observational data, which can be confounded and negatively affect the performance of intervention response estimators used for counterfactual inference. Secondly, the continuity of the dose prevents the adoption of approaches used to estimate responses to binary-valued interventions. That is why the machine learning (ML) community has proposed several tailored CADR estimators. Yet, the proposal of most of these methods requires strong assumptions on the distribution of data and the assignment of interventions, which go beyond the standard assumptions in causal inference. Whereas previous works have so far focused on smooth shifts in covariate distributions across doses, in this work, we will study estimating CADR from clustered data and where different doses are assigned to different segments of a population. On a novel benchmarking dataset, we show the impacts of clustered data on model performance. Additionally, we propose an estimator, CBRNet, that enables the application of representation balancing for CADR estimation through clustering the covariate space and a novel loss function. CBRNet learns cluster-agnostic and hence dose-agnostic covariate representations for unbiased CADR inference. We run extensive experiments to illustrate the workings of our method and compare it with the state of the art in ML for CADR estimation.

## 1 Introduction

Predicting responses to interventions conditional on a unit's covariates, so-called "conditional-average intervention responses" has become a popular field of machine learning (ML) research (Shalit et al., 2017; Shi

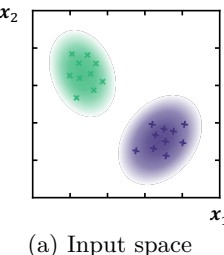
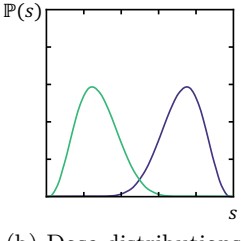
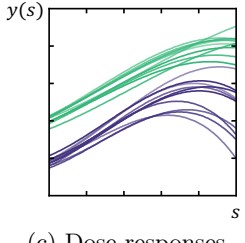

(a) Input space       (b) Dose distributions       (c) Dose responses

Figure 1: **Illustration of confounding by cluster (CBC).** (a) We assume that there are clusters of similar units in the data, visualized by the green and blue point cloud in an exemplary two-dimensional covariate space. (b) The dose assignment mechanism is a function of the cluster so similar units are assigned similar doses. This results in different distributions of dose assignment per cluster. (c) As dose responses ($y(s)$) are heterogeneous and depend on a unit's covariates ($\mathbf{x}_1$, $\mathbf{x}_2$), the dose response curves of units in different clusters (green and blue curves) are different, resulting in confounded data.

et al., 2019).[1] While a large share of studies focuses on estimating responses to categorical and especially binary-valued interventions (Shalit et al., 2017; Yoon et al., 2018; Shi et al., 2019; Johansson et al., 2022), less focus has been put on continuous-valued interventions, so interventions with an associated intensity or dose (Schwab et al., 2019). Yet, estimating the conditional-average "dose response" (CADR) is of great interest in many domains of applications, such as marketing (Farrelly et al., 2005), medicine (Calabrese, 2016), or education (Turk, 2019), especially when too much or too little intervention can have negative effects (Frei & Canellos, 1980), or when decision-makers must optimize resource usage (Vanderschueren et al., 2023).

Such responses must often be estimated from observational data. Hence, next to high-dimensional covariates and non-linear dose responses, an important challenge is for an estimator to adjust for confounding (Bica et al., 2020), so the presence of a covariate vector that influences the dose assignment, as well as the response (Haneuse, 2016). Not accounting for confounding might hinder methods from learning unbiased estimates of CADR by overfitting the dose assignment mechanism (Shalit et al., 2017). Many approaches to estimating responses to binary-valued interventions under observed confounding, however, cannot easily be transferred to the dose response setting (cf. Section 2). This is why the ML community has proposed several tailored methods to learn CADR from observational data (Schwab et al., 2019; Bica et al., 2020; Nie et al., 2021; Wang et al., 2022). Next to the standard assumptions necessary for causal inference (Stone, 1993), most of these estimators rely on further assumptions on the underlying data, such as on the distribution of covariates and the assignment of interventions and doses (Nie et al., 2021; Wang et al., 2022). These assumptions drive algorithm development, yet are often not motivated by investigations of real-life scenarios and their accompanying data-generating processes (DGPs). This potentially explains why ML estimators for intervention response estimation have seen little adoption in practice (Curth et al., 2021). One such assumption is a smooth shift in covariate distributions across different doses, as required by Wang et al. (2022) and satisfied in the synthetic benchmarking datasets by Bica et al. (2020) and Nie et al. (2021) which are frequently used in the literature.

In this paper, we investigate CADR estimation from data in which such assumptions are violated. Especially, we will investigate scenarios in which units are clustered and where different doses are assigned to different segments of a population, leading to "confounding by cluster" (CBC; cf. Figure 1). Such data is prevalent in many domains, such as loan and insurance pricing, where groups of similar customers receive similar prices (Phillips, 2013), or epidemiology, where the impacts of clustered data on modeling have been studied widely (Berlin et al., 1999; Seaman et al., 2014). In ML research, however, we find that little interest has been paid to understanding the impact of clustered data on intervention and dose response estimates. In this paper, we take a step towards closing this gap in the literature. By analyzing the real-world applications of CADR estimation in healthcare, public policy, and business, we abstract DGPs for clustered data and create

---

[1]Some literature also refers to the conditional-average response as the "individual" response. For a discussion on why the term "conditional-average" is preferred over the term "individual", we refer the reader to Vegetabile (2021).

a novel semi-synthetic benchmarking dataset for dose response estimation from clustered data with varying levels of confounding. We evaluate several ML methods, including traditional supervised learning methods and tailored ML methods for CADR estimation. In addition, we propose a new method for estimating CADR from clustered observational data with cluster-based dose assignment. Our method is called CBRNet (**C**luster-robust dose response estimation through **B**alanced **R**epresentation learning with neural **Net**works, cf. Figure 3) and is motivated by preceding studies on estimating intervention responses under confounding through the balancing of covariate distributions (Shalit et al., 2017; Schwab et al., 2018).

**Contributions.**  Our paper makes three important contributions to the ML literature on dose response estimation: (1) To the best of our knowledge, we are the first to investigate the impacts of learning from clustered data on the performance of ML estimators for CADR; (2) we propose a novel semi-synthetic benchmark that simulates CBC in observational data and that allows for testing its impacts on estimators; (3) we introduce CBRNet, a method for CADR estimation from clustered data. By introducing a novel loss function and by clustering the covariate space, we present an intuitive and effective approach to use representation balancing for CADR estimation. We perform extensive experiments to test the performance of CBRNet, both on our newly presented benchmark and on previously established datasets.

**Outline.**  Section 2 discusses related work on ML methods for CADR estimation and preceding benchmarking practices. Our problem formulation follows in Section 3. We present our method, CBRNet, in Section 4, discussing motivation and architecture. Our experimental setup is presented in Section 5, with results following in Section 6. We conclude with Section 7, discussing the impact of our work and future research directions.

## 2 Context

**Background.**  The majority of ML literature on estimating intervention responses has been concerned with "treatment effects", so the differences in outcome between applying a specific intervention, the "treatment", or not (Johansson et al., 2016; Shalit et al., 2017; Louizos et al., 2017; Nie & Wager, 2021; Wager & Athey, 2018; Shi et al., 2019). Less attention has been paid to settings with interventions that are continuous-valued, so that have an associated intensity or dose. Nevertheless, estimating continuous-valued intervention responses, or simply "dose responses", is relevant in a plethora of domains of application (Farrelly et al., 2005; Calabrese, 2016; Turk, 2019; Vanderschueren et al., 2023). Compared to estimating treatment effects, dose response estimation comes with unique challenges (Schwab et al., 2019), which make this an interesting and relevant field of research.

For estimating treatment effects, controlled experiments, like randomized controlled trials (RCTs; Deaton & Cartwright, 2018), are often considered the gold standard. Yet, due to the practically infinite amount of possible doses, their application is signficantly complicated for dose response estimation (Holland-Letz & Kopp-Schneider, 2014). Even when a suitable number of units is available, such experiments might be prohibitively costly.

Alternatively, dose responses can be estimated using observational data. Yet, as with treatment effect estimation, this comes with several challenges, most importantly: a) The impossibility of observing counterfactual outcomes (Holland, 1986) and b) the presence of a dose assignment mechanism, which may have led to confounding in the data (Varadhan & Seeger, 2013). These challenges may prevent the adoption of traditional supervised learning methods (Schwab et al., 2019).

**ML dose response estimators.**  Phenomena, like observed confounding in data, are well described and widely researched for treatment effect estimation (Curth et al., 2021). However, many methods presented for estimating treatment effects do not translate to the dose response setting. Methods such as matching (Ho et al., 2007), the propensity score (Rosenbaum & Rubin, 1983), representation balancing (Johansson et al., 2016), representation learning (Louizos et al., 2017) and tailored tree-based estimators (Hill, 2011; Wager & Athey, 2018) rely on the presence of two distinct groups, such as treated units and control units.

Hence, several tailored ML estimators for dose responses were proposed. Hirano & Imbens (2004) propose the Hirano-Imbens estimator (HIE), built on the idea of the generalized propensity score (GPS), extending the propensity score to continuous-valued interventions. Schwab et al. (2019) propose DRNet, a neural architecture to learn conditional-average responses to multiple intervention options with an associated dose. Their method can be combined with several balancing techniques to tackle the confounding of intervention assignment, yet not the confounding of assigned doses. Alternatively, Bica et al. (2020) propose SCIGAN, which leverages generative adversarial networks (GANs; Goodfellow et al., 2020), a method that learns how to generate counterfactual responses. Generating counterfactual responses can remove confounding in the data, enabling learning the dose-response via supervised learning methods. Nie et al. (2021) propose VCNet, a generalization of DRNet that accounts for the continuity of dose responses, by training a varying coefficient network (Hastie & Tibshirani, 1993). Nie et al. (2021) study the estimation of average dose responses, by combining estimates of the conditional-average response through functional targeted regularization, a method to build doubly-robust average dose response estimators inspired by van der Laan & Rubin (2006). Wang et al. (2022) also tackle estimating the average dose response. Their estimator, ADMIT, minimizes the maximal distributional difference between units of discrete dose intervals, assuming a smooth shift in the probability distribution of covariates for units of different doses. Zhang et al. (2023) propose TransTEE, a method using the transformer architecture (Vaswani et al., 2017) to build intervention effect estimators. They claim that the attention mechanisms in these are superior in modeling observed confounders. Bellot et al. (2022) propose an architecture similar to DRNet, which uses multiple prediction heads for different interventions with a dose, and use representation balancing to overcome intervention confounding for conditional-average response estimation. Dose confounding is not addressed in their work.

Note that, even though methods such as VCNet and ADMIT were proposed for average dose response estimation, their methodologies can be used to estimate conditional-average responses. Specifically, any preceding method for average response estimation first trains a conditional-average response estimator. A simple estimator of the average response can be obtained by the law of total expectation and averaging the conditional-average estimates over a population of interest (Abrevaya et al., 2015). For a detailed description of our problem setup, see Section 3.

**Benchmarking practices in ML for dose response estimation.** As counterfactual outcomes cannot be observed, ML estimators for intervention responses cannot easily be evaluated using observational data (Schwab et al., 2019; Bica et al., 2020). This is different from, for example, supervised learning (Hastie & Tibshirani, 1993). Alternatively, many researchers derive theoretical guarantees for their estimators' performance based on assumptions or use semi- or fully synthetic datasets for an empirical evaluation. Neither of these two approaches ensures good real-life performance. Datasets are usually constructed according to a synthetic data-generating process (DGP), which defines causal relationships between covariates, interventions, and outcomes. Prominent examples of such datasets for dose response estimation include the ones presented by Bica et al. (2020) and Nie et al. (2021).

Curth et al. (2021) show that most research lacks analysis of the aspects of a synthetic dataset, and that challenges in a dataset are often incompletely understood. In contrast to the preceding works, we argue that the creation of dose response estimators should be use-case driven and that we need a better understanding of the effects of different DGPs on model performance. Hence, we base the methodology of this work on abstractions of real-life use cases of CADR estimation, motivating the creation of our new method.

## 3 Problem formulation

We leverage the Neyman-Rubin potential outcomes framework (Splawa-Neyman et al., 1990; Rubin, 1974) and assume that we observe units defined by a pre-intervention covariate vector $\mathbf{X} \in \mathcal{X} \subset \mathbb{R}^m$ in some $m$-dimensional feature space. Every unit is assigned a positive dose $S \in \mathcal{S} \subset \mathbb{R}^+$. For the remainder of our paper and without loss of generality, we set $\mathcal{S} = [0, 1]$ to be the unit interval. Every unit has potential responses, or "outcomes" $Y(s) \in \mathcal{Y} \subset \mathbb{R}$, which are the responses had it received the dose $s$. Following Schwab et al. (2019), we call $Y(s)$ the "dose response". The expected response for a dose given a unit's

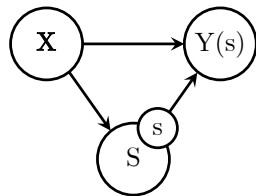

Figure 2: **Single world intervention graph** (SWIG) illustrating causal dependencies between variables in the training data. Our goal is to estimate the response of a unit to a dose $s$.

---

covariates, the "conditional-average dose response" (CADR), is subsequently given by

$$\mu(s, \mathbf{x}) = \mathbb{E}[Y(s)|\mathbf{X} = \mathbf{x}]. \tag{1}$$

Our goal is to estimate the CADR from observational data. We expect to have access to data in the form of $\mathcal{D}_n = \{(\mathbf{x}_i, s_i, y_i)\}_{i=1}^n$, where $n$ is the total number of observed units. $\mathbf{x}_i$, $s_i$, and $y_i$ are the covariates, the assigned dose, and the observed response of the $i$-th unit respectively. We also refer to $y_i$ as the "factual response". We operate under the standard fundamental problem of causal inference (Holland, 1986), so responses of a unit $i$ to doses different from $s_i$, so-called "counterfactual responses", are never observed.

One challenge in CADR estimation from observational data lies in the relationships between variables $\mathbf{X}$, $S$, and $Y$. In observational data, unlike in controlled environments, doses were likely assigned according to some (partially) unknown assignment mechanism based on units' covariates. In the case of heterogenuous dose responses, such a mechanism might introduce confounding, in which the covariate vector influences both dose assignment and the respective dose response. We illustrate the relationships between the different variables in the data in a single-world intervention graph (SWIG; Richardson & Robins, 2013) in Figure 2. To learn an unbiased estimate of the CADR, a method must hence adjust for any potential confounding (Shalit et al., 2017).

Finding unbiased estimates of intervention responses relies on a set of untestable assumptions (Stone, 1993). In line with preceding works, we require the following standard assumptions to hold:

**Assumption 1.** *(Consistency) The observed outcome $Y_i$ for a unit i that was assigned dose $s_i$ is the potential outcome $Y_i(s_i)$.*

**Assumption 2.** *(No hidden confounders) The assigned dose $S$ is conditionally independent of the potential outcome $Y(s)$ given the covariates $\mathbf{X}$, so $\{Y(s)|s \in \mathcal{S}\} \perp\!\!\!\perp S|\mathbf{X}$*

**Assumption 3.** *(Overlap) Every unit has a greater-than-zero probability of receiving any dose, so $\forall s \in \mathcal{S} : \forall \mathbf{x} \in \mathcal{X}$ with $\mathbb{P}(\mathbf{x}) > 0 : 0 < \mathbb{P}(s|\mathbf{x}) < 1$*

*Note:* As indicated in Section 2, estimating the CADR is also relevant for estimating average dose responses (ADR, $\hat{\mu}(s)$). Methods for ADR estimation are typically first training a CADR estimator $\hat{\mu}(d, \mathbf{x})$ and are subsequently deriving an estimate of the ADR through the law of total expectation by calculating $\hat{\mu}(s) = \frac{1}{n}\sum_{i=1}^n \hat{\mu}(s, \mathbf{x}_i)$.

## 4 CBRNet: A method to tackle confounding by cluster in CADR estimation

**Motivation.** It is widely understood that use cases and applications should drive method selection (Luo, 2016), and that algorithm architecture plays a major role in estimator performance (Alaa & van der Schaar, 2018). This is especially true for intervention response estimators, whose performance cannot be studied on real data. We hence argue that algorithm development should be aligned more closely with the study of real-world applications. By now, the ML community has relied mostly on a small set of benchmarking datasets (Curth et al., 2021), such as the IHDP dataset presented by Hill (2011) for treatment effect estimation, or

the datasets by Bica et al. (2020) and Nie et al. (2021) for dose response estimation. While posing distinct challenges to estimators and relying on covariates acquired from real-world studies, these datasets are created using synthetic DGPs specifying intervention assignment and potential outcomes. These mechanisms are often created without further motivation, raising questions about their realisticness.

For this manuscript, we take an alternative approach and will focus on CADR estimation from clustered data with cluster-based dose assignment. Whereas in preceding works intervention and dose assignment are typically specific to a unit's covariates, we assume that similar units form clusters and that dose assignment is conditional on the cluster (cf. Figure 1). So, every unit $i$ is assigned a cluster $c_i$ according to some clustering function $g(\mathbf{x}) : \mathbf{X} \to C$. If that assumption holds, we can reformulate "No hidden confounders" (cf. Section 3) as $\{Y(s)|s \in \mathcal{S}\} \perp\!\!\!\perp S|C$. While not previously studied in the ML literature on dose response estimation,

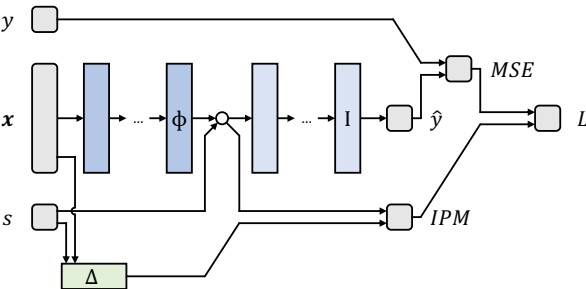

Figure 3: **Architecture of CBRNet.** The network consists of three parts. A representation learner $\Phi$, an inference network $I$, and a clustering function $\Delta$. To overcome confounding by cluster, $\Phi$ is trained to learn a dose-agnostic representation by minimizing a tailored integral probability metric (IPM) over the response space given the clusters identified by $\Delta$. Subsequently, the inference network $I$ is trained to learn the response of a unit to a dose by minimizing a standard mean squared error loss (MSE). For a full description of our method see Section 4.

such mechanisms occur in several domains of application. Assuming that dose responses are heterogeneous, such cluster-based assignment can introduce confounding in the data, often referred to as "confounding by cluster" (Localio et al., 2002; Seaman et al., 2014; Zetterqvist et al., 2016; Berlin et al., 1999). We identified several real-life scenarios in which such mechanisms apply:

- In the pricing of loan and insurance products, companies might want to identify the effect of price on a customer's willingness to buy or renew a product, yet in accordance with established business processes similar customers were assigned to different price tiers (Yeo et al., 2002; Phillips, 2013);

- in marketing, a company might want to predict the effect of exposure to a marketing campaign, yet customers were grouped by loyalty programs and have hence received distinct amounts of discounts (Jonker et al., 2004; Ho et al., 2012);

- in medicine, practitioners might want to understand the effect of varying amounts of a drug on health outcomes, yet patients were grouped by biomarkers to receive certain doses (Schacht et al., 2014; Verhaart et al., 2014);

- in education, one might want to predict the effectiveness of certain learning programs on students' educational success, yet students were divided based on ability, leaving them exposed to similar interventions (Betts & Shkolnik, 2000);

- in public policy, policymakers might want to estimate the effects of subsidies on research organizations, yet organizations often form clusters with similar characteristics, which apply for grants jointly (Broekel et al., 2015).

In all of the above scenarios, the data will likely be confounded. To learn an unbiased estimate of the CADR, we must find an estimator robust to such observed confounding. As doses are driven by the cluster, supervised estimators that do not address confounding or treatment assignment bias might overfit the assignment mechanism and learn biased estimates of dose responses. Feed-forward neural networks, for example, tend to memorize the relation between the dose and the outcome as observed in the available data (Schwab et al., 2019). However, due to the non-random treatment assignment mechanism, some clusters will be over- or underrepresented at a certain dose, compared to their proportion in the overall population. These "traditional" supervised methods do not account for confounding, resulting from the dose assignment mechanism (cf. below). Therefore, the resulting models may not generalize well to units of different clusters, with different characteristics and different responses to doses. As illustrated in Figure 1c, the effect of a

particular dose on the outcome is different for observations from a different cluster. We propose to solve this challenge by building a model that learns cluster-agnostic, and hence dose-agnostic representations of a unit's covariates, for unbiased CADR estimation.

**Architecture.** The architecture of CBRNet is shown in Figure 3. CBRNet consists of three parts $\Phi$, $\Delta$, and $I$, based on our motivation in the paragraph above. $\Phi : \mathcal{X} \to \mathcal{R}$ is a representation learning function mapping the covariates into the representation space $\mathcal{R} \subset \mathbb{R}^n$ to learn a cluster-agnostic representation. We use a standard feed-forward neural network for $\Phi$, where both the number of layers and the number of hidden nodes per layer are hyperparameters. $\Delta : \mathcal{X} \times \mathcal{S} \to \{1, \ldots, k\}$ is a clustering function mapping a unit to one of $k$ clusters by taking as input the covariates and doses of a unit. We assume that dose assignment is conditional on the cluster, hence adding the dose as input to the clustering function is expected to improve performance in separating clusters with heterogeneous doses. For an overview of possible clustering algorithms, see Madhulatha (2012). We propose to use $k$-means (Lloyd, 1982) as a clustering function, minimizing Euclidian distances between units of a certain cluster. It is widely adopted in business and beyond (Wu, 2012) and has previously been used in treatment effect estimation (Berrevoets et al., 2020). We train $\Delta$ on all available training data and do not alter it during the training of the remaining network components (see paragraph below). $I : \mathcal{R} \times S \to \mathbb{R}$ is an inference function taking as input the covariates in representation space $\mathcal{R}$ and a dose, to learn the CADR. As for $\Phi$, we propose $I$ to be a feed-forward neural network with flexible hyperparameters, as similarly adopted by Shalit et al. (2017), Schwab et al. (2019), and Bica et al. (2020).

**Model selection and training.** CBRNet is trained using gradient descent over the training dataset $\mathcal{D}_n = \{(\mathbf{x}_i, s_i, y_i)\}_{i=1}^n$ minimizing loss $L$ defined as

$$L(\mathbf{x}, s, y) = L_I(y, \hat{y}) + \lambda * L_\Phi(\Phi, \Delta, \mathcal{D}). \tag{2}$$

$L_I$ is a standard mean squared error (MSE) loss

$$L_I = MSE(y, \hat{y}) = \frac{1}{N} \sum_{i=1}^{N} (y_i - \hat{y}_i)^2 \tag{3}$$

where $y_i$ is the true outcome of unit $i$ and $\hat{y}_i$ is the estimated outcome as calculated by the inference function $I$. $L_\Phi$ is a loss that ensures that $\Phi$ learns a cluster-agnostic representation. To achieve this, we use integral probability metrics (IPMs), which have been previously used to improve ML models for treatment effect estimation by minimizing distances between treated and untreated observations in the representation space (Shalit et al., 2017). Metrics, such as the maximum mean discrepancy (MMD; Gretton et al., 2006) or the Wasserstein distance (Kantorovich, 1960; Vaserstein, 1969), are defined for strictly two distributions. To enable calculating distances between the distributions of different clusters in the representation space $\mathcal{R}$, we calculate $L_\Phi$ as

$$L_\Phi(\Phi, \Delta, \mathcal{D}) = \frac{1}{k-1} \sum_{i=1}^{k-1} IPM\left(\{\Phi(\mathbf{x}_j)\}_{j:\Delta(\mathbf{x}_j,s_j)=i}, \{\Phi(\mathbf{x}_j)\}_{j:\Delta(\mathbf{x}_j,s_j)=k}\right). \tag{4}$$

Intuitively, we choose one base cluster $k$ and take the average over the pair-wise distances between the remaining clusters and the base cluster as regularization loss. This approach is inspired by the generalization of the PEHE metric (Precision in Estimating Heterogeneous Effects, Hill, 2011) to multiple treatments as proposed by Schwab et al. (2018). $\lambda$ is a hyperparameter to balance the two losses, $L_I$ and $L_\Phi$. Our proposition of the regularization loss allows for a flexible choice of IPM and cluster number $k$, such as linear or kernel MMDs or the Wasserstein distance. We will test the impact of different choices in our empirical evaluation in the following sections. In practice, we choose as base $k$ the cluster with the most observed units, in order to ensure accurate empirical approximation of the IPMs.

As discussed in previous works on ML for CADR estimators, model selection and hyperparameter tuning are inherently difficult due to the unavailability of counterfactual outcomes (Schwab et al., 2019; Bica et al., 2020). For an overview, Curth & Van Der Schaar (2023) discuss several approaches to model selection.

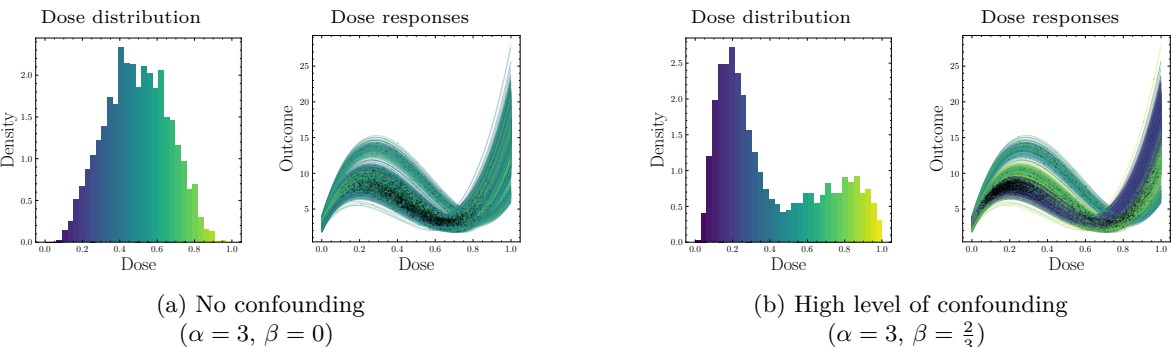

(a) No confounding
($\alpha = 3$, $\beta = 0$)

(b) High level of confounding
($\alpha = 3$, $\beta = \frac{2}{3}$)

Figure 4: **Visualization of level on confounding in dry bean dataset.** Per subfigure, the left plot visualizes the dose distribution across the population and provides a color legend. The right plot visualizes the dose response space. Per dose response, the line color corresponds to the assigned dose, which is marked with a dot ($\cdot$). In the unconfounded case (a), doses are homogeneously distributed across units. In the confounded case (b), doses are assigned conditional on clusters, as evident by the color separation of different dose responses.

Choosing one procedure over another can significantly impact response estimation and model bias. For our empirical evaluation and to not confuse the performance of CADR estimators with the performance of model selection procedures, we propose to use a standard mean squared error over observed units in a validation set to optimize hyperparameters of CBRNet. For further detail, we refer to Section 5 and our technical implementation in Appendix E. We propose to use a validation set to optimize hyperparameters and to choose the best model in terms of the validation set MSE.

## 5   Experimental Evaluation

We evaluate CBRNet empirically by comparing it to several benchmarking methods on a novel semi-synthetic dataset, the "Dry bean-DR data". The following paragraphs will discuss the creation of this data, the benchmarking methods, and the metrics used for evaluation. The dataset is available publically and has been provided with the code for this manuscript. We want to mention that we do not claim the Dry bean-DR dataset to be a superior dataset for evaluating CADR estimators. Instead, our proposal of the dataset is meant to complement the set of available benchmarking datasets. The key novelty compared to other established datasets is the cluster-based dose assignment.

**Data generation.**   To simulate clustered data and cluster-based intervention assignments, we use a semi-synthetic setup. While previous works primarily motivate the realisticness of their data by the use of covariates sampled from real-life experiments, such as the TCGA data (Cancer Genome Atlas Research Network et al., 2013), a dataset on gene sequences of cancer cells, we put a strong emphasis on the intervention assignment mechanisms as a prime factor driving confounding. Our data generation starts by taking the covariates of the dry bean dataset (Koklu & Ozkan, 2020). The dataset contains 13,611 samples of seven different types of beans, listing 16 unique features per sample derived from a computer vision analysis. We relate this data to intervention response estimation by considering a situation in which we want to understand the effect of different irrigation levels on crop yields (Goldstein et al., 2017; Dehghanisanij et al., 2022). Understanding the conditional-average responses to irrigation could lead to improved yields, yet we have to estimate these responses from observational data, stemming from established farming practices. Hence, the data likely is confounded. The data generation now proceeds in three steps:

*Step 1 (Clustering):* We take as clusters the types of beans in the original data and randomly aggregate them to form three clusters. For a unit $i$, we denote the corresponding cluster as $c_i \in \{1, 2, 3\}$. During model training and application, we expect the cluster affiliation to be unknown.

*Step 2 (Dose assignment):* Different from preceding datasets (Schwab et al., 2019; Bica et al., 2020; Nie et al., 2021), we assign doses based on clusters which will lead to a more rapid shift in covariate distributions across different dose levels in the data. We refer back to Section 4 for real-world examples of such mechanisms. Every cluster $j$ is assigned a different "modal dose" $m_j$, which is randomly drawn from $\{\frac{1-\beta}{2}, \frac{1}{2}, \frac{1+\beta}{2}\}$ without replacement. The parameter $\beta \in [0, 1]$ determines the variability of doses across clusters. For $\beta = 0$ the dose assignment is homogenous across clusters. For $\beta = 1$, dose assignment is maximally heterogeneous. We set $\beta = \frac{1}{2}$ for our main experiments, and show the impact of different levels of $\beta$ in Appendix C. As the cluster of a unit is determined by its location in the covariate space, the dose is implicitly dependent on the covariates.

To add variability of doses within clusters, per unit $i$, we sample the individual dose $s_i$ from a Beta-distribution

$$S_i \sim \text{Beta}(\alpha, \frac{\alpha}{m_{c_i}} + (1 - \alpha)), \tag{5}$$

as motivated by Bica et al. (2020), ensuring that the mode of $S_i$ is $m_{c_i}$. The parameter $\alpha$ determines the variance of the Beta-distribution. For $\alpha = 0$, the dose of a unit is sampled from a uniform distribution, leading to no confounding, as the dose is unconditional on a unit's covariates. For $\alpha \to \infty$, the dose assignment is fully deterministic, and every unit in a cluster will be assigned the model dose. We can think of $\alpha$ as the confounding strength. We visualize the impacts of parameters $\alpha$ and $\beta$ in Appendix A.

*Step 3 (Response calculation):* We define a ground truth dose response per unit, which allows for the calculation of counterfactual outcomes. The conditional-average dose response for a covariate vector $\mathbf{x}_i$ and a dose $s$ is defined as

$$\mu(s, \mathbf{x}_i) = 10 \left( \mathbf{w}_1^\mathsf{T} \mathbf{x} + 12s \left( s - \frac{3}{4} \frac{\mathbf{w}_2^\mathsf{T} \mathbf{x}}{\mathbf{w}_3^\mathsf{T} \mathbf{x}} \right)^2 \right), \tag{6}$$

where $\mathbf{w}_l \in \mathbb{R}^{16}$ for $l \in \{1, 2, 3\}$ is a weight vector. We randomly sample half of the weights from a uniform distribution $\mathcal{U}(0, 1)$ and set the remaining weights to zero to add noise covariates (Shi et al., 2019). We test the effect of different degrees of sparseness in the weight vector in Appendix D. The individual dose response per unit is finally calculated as $y_i = \mu(s_i, \mathbf{x}_i) + \epsilon$ where $\epsilon \sim \mathcal{N}(0, 1)$ is a random error term. The final data for different values of $\alpha$ and $\beta$ is visualized in Figure 4 illustrating the confounding.

**Benchmarks.** We compare CBRNet against several relevant baselines. First, we compare it against supervised learning algorithms, namely linear regression, a regression tree, xgboost, as a state-of-the-art implementation of gradient-boosted trees and a feedforward multi-layer perceptron (MLP). Second, we compare it against DRNet (Schwab et al., 2019) and VCNet (Nie et al., 2021), two popular and competitive dose response estimators. For a discussion of these methods, see Section 2. To understand the impacts of different IPMs on model performance, we train CBRNet using the linear MMD ($\text{MMD}_{lin}$), a kernel MMD ($\text{MMD}_{rbf}$) using the radial basis function kernel, and the Wasserstein distance.

**Performance metrics.** We evaluate the performance of each method using the MISE metric introduced by Schwab et al. (2019), which measures the ability of a method to estimate the CADR over all units in the test data and all dose levels. For a test dataset with $n$ units, the MISE is defined as

$$\text{MISE} = \frac{1}{n} \sum_{i=1}^{n} \int_{\mathcal{S}} \left( \mu(s, \mathbf{x}_i) - \hat{\mu}(s, \mathbf{x}_i) \right)^2 \mathrm{d}s. \tag{7}$$

## 6 Empirical Results

**Performance on the dry bean dataset.** For a total of 17 different combinations of $\alpha$ and $\beta$, we generate 10 random instances of the Dry bean-DR data. We use 70% of the data for training, 10% as a validation set for hyperparameter tuning, and 20% as a test set for calculating performance metrics.

Fixing $\beta = \frac{1}{2}$, we report the performance of all nine estimators for different levels of $\alpha$ in Table 1. Performance under different levels of $\beta$ is stated in our extended results section in Appendix C. CBRNet outperforms all

Table 1: **MISE per method on dry bean dataset** for $\beta = \frac{1}{2}$ and varying levels of $\alpha$. We compare CBRNet against several benchmarking methods over varying levels of confounding by cluster. The best results are printed in **bold**, second-best results are in *italics*. Results for different values of $\alpha$ can be found in Appendix C.

$\beta = \frac{1}{2}$

| Model | $\alpha$ | | | |
| --- | --- | --- | --- | --- |
| | 1.0 | 2.0 | 3.0 | 4.0 |
| Linear Regression | $3.32 \pm 0.03$ | $3.40 \pm 0.06$ | $3.30 \pm 0.02$ | $3.32 \pm 0.03$ |
| CART | $1.31 \pm 0.06$ | $1.40 \pm 0.09$ | $1.56 \pm 0.15$ | $1.53 \pm 0.13$ |
| xgboost | $0.91 \pm 0.07$ | $1.16 \pm 0.09$ | $1.15 \pm 0.07$ | $1.28 \pm 0.04$ |
| MLP | $0.54 \pm 0.04$ | $0.67 \pm 0.08$ | $0.83 \pm 0.09$ | $1.06 \pm 0.08$ |
| DRNet | $0.62 \pm 0.03$ | $0.78 \pm 0.13$ | $0.82 \pm 0.07$ | $0.98 \pm 0.06$ |
| VCNet | $0.75 \pm 0.11$ | $1.07 \pm 0.10$ | $1.18 \pm 0.11$ | $1.35 \pm 0.07$ |
| CBRNet(MMD$_{lin}$) | *0.44* $\pm 0.09$ | *0.47* $\pm 0.12$ | **0.44** $\pm 0.07$ | *0.63* $\pm 0.15$ |
| CBRNet(MMD$_{rbf}$) | $0.45 \pm 0.10$ | $0.53 \pm 0.14$ | $0.50 \pm 0.10$ | $0.71 \pm 0.17$ |
| CBRNet(Wass) | **0.43** $\pm 0.09$ | **0.46** $\pm 0.10$ | *0.45* $\pm 0.08$ | **0.55** $\pm 0.11$ |

benchmark methods across different levels of confounding, indicating that our proposed model architecture is well-suited to learn CADR from clustered data.

CBRNet shows higher robustness to confounding by cluster than its benchmarks when the confounding strength $\alpha$ increases. Irrespective of the chosen IPM for regularization, the performance of CBRNet is stable for low to moderate levels of $\alpha$, only increasing for the highest value of $\alpha = 4$. For all other models, performance is strictly decreasing for increases in $\alpha$.

Similar behavior can be observed for different values of $\beta$. A notable case is given for $\beta = 0$, hence when there is no variability of doses between clusters (cf. Table 2 in Appendix C). For $\beta = 0$, the data is unconfounded, as doses are sampled from the same distribution for all units and all clusters. So, CBRNet also outperforms the benchmark methods when learning CADR from clustered, but unconfounded data.

**Hyperparameter sensitivity** We further investigate the sources of gain of our proposed architecture by testing the impact of two critical hyperparameters: (a) The strength of the regularization $\lambda$, and (b) the number of clusters $k$ identified by the $k$-means clustering. The results of those experiments are summarized in Figure 5. For both experiments, we fix the hyperparameters of CBRNet and iterate over varying levels of $\lambda$ and $k$. To stabilize results, for every parameter configuration, we generate 10 random instances of the Dry bean-DR data with $\alpha = \frac{2}{3}$ and $\beta = 3$.

In Appendix B we visualize the impact of the IMP regularization on the representation space. We find a strictly positive impact on model performance for low to moderate levels of regularization ($\lambda < 0.1$). These results support the positive impact of regularizing for differences in cluster distributions in the representation space. These results are also independent of the chosen IPM, with MMDs and Wasserstein distance performing comparably. Only for an increased level of regularization, does the predictive performance of CBRNet decrease. This indicates that an overregularization is possible

The performance of CBRNet appears to be robust to the specified number of clusters $k$, with performance converging on the Dry bean-DR data for larger $k$. We hence expect our method to have favorable asymptotical properties for unclustered data, as long as the dose assignment can be approximated sufficiently well by a large enough number of clusters and if sufficient data is available to approximate the IPMs. In applications, next to before-mentioned model selection procedures, the number of clusters can be selected together with domain experts, or by visual analysis.

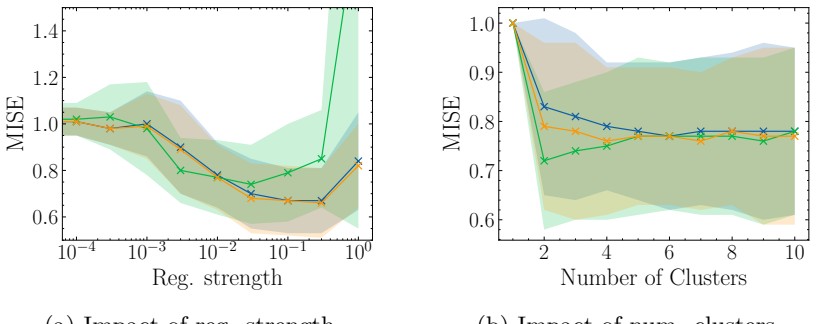

(a) Impact of reg. strength    (b) Impact of num. clusters

Figure 5: **Hyperparameter robustness.** (a) Moderate levels of regularization improve model performance. Regularizing by the MMD with a radial basis function kernel (rbf) appears to overregularize earlier than for linear MMD and Wasserstein distance. (b) We see robustness to the number of clusters. Performance converges for all IPMs.

**Performance on established datasets**  To judge the performance of CBRNet on unclustered data, we run experiments on previously established benchmarking datasets proposed by Bica et al. (2020)[2] and Nie et al. (2021). CBRNet performs competitively, beating all of its benchmark methods on three of the four datasets. However, further decomposition is needed to understand the precise challenges in these datasets to attribute performance to either the general architecture of CBRNet, or the IPM regularization. We provide the full results on these datasets in Table 5 in Appendix C. While good performance on a benchmarking dataset is no guarantee for performance in a real-world application, we consider these results to indicate the effectiveness of our newly developed approach.

## 7  Conclusion

We studied the estimation of conditional-average dose responses (CADR) from clustered observational data. Practitioners face clustered data in several real-life scenarios, covering applications in, e.g., business, economics, and healthcare (cf. Section 4). In such scenarios, dose assignment is often cluster-specific, potentially leading to confounding. This confounding mechanism is distinct from scenarios previously studied in the ML literature on intervention response estimation and has impacts on the validity of assumptions that are often made beyond standard ones necessary for applying causal inference (cf. Section 2).

To enable thorough testing of estimators under the presence of such data, we created a novel dataset, the "Dry bean-DR data", based on the covariates initially presented by Koklu & Ozkan (2020). Experiments on this dataset reveal that traditional supervised estimators and ML estimators tailored to CADR estimation suffer from confounding by cluster. To work towards solving this problem, we proposed a new ML estimator, CBRNet. Using a novel loss function that averages integral probability metrics over any number of clusters in the covariates space, we can leverage representation balancing for CADR estimation and prevent confounding biases.

In comparison with its benchmarks, CBRNet shows improved performance, indicating our approach's appropriateness. Further experiments using established datasets revealed that CBRNet can similarly achieve competitive performance on unclustered data.

Our research has shown that understanding the context and unique characteristics of a CADR estimation problem is imperative. Variations in the underlying data-generating process of observational data can have adverse effects on model performance. To ensure good performance, especially given the inherent difficulty of model selection in causal inference, practitioners must check the validity of any method-specific assumptions, for example, by involving domain experts (Pearl, 2022).

---

[2]The DGP proposed by Bica et al. (2020) can incorporate up to three different interventions, each with an associated dose. To comply with the architecture of CBRNet, we generate responses to only a single intervention.

Our implementation of CBRNet is available online for practitioners and fellow researchers to build upon (cf. Appendix E). Our implementation uses a basic `scikit-learn` syntax (Pedregosa et al., 2011), enabling efficient model setup, training and inference:

```
1  from src.methods.neural import CBRNet
2  model = CBRNet()
3  model.fit(X_train, Y_train, S_train)
4  model.predict(X_test, S_test)
```

All relevant hyperparameters can be specified. The architecture of our method was motivated by the analysis of real-world scenarios, leveraging and extending the concept of representation balancing, first proposed in the literature on ML for causal inference by Shalit et al. (2017). We highlight three potential future research directions: (1) A theoretical analysis of CBRNet and asymptotical guarantees of model performance in clustered and unclustered data. Especially, an analysis of the necessary amount of data for effective regularization could benefit the adoption of our model, as the IPMs for regularization are approximated empirically. (2) Further research on model selection and hyperparameter tuning. While using a validation set MSE can be a practical first guess (Curth & Van Der Schaar, 2023), alternative approaches could further add performance guarantees when training models on real-life data. This research is not specific to CBRNet, but is relevant to the wider related literature. (3) Finally, our research is specific to interventions with an associated dose, hence extending our experiments to binary-valued interventions could be a promising and insightful direction, especially given the limited number of benchmarking datasets for treatment effect estimation (Curth et al., 2021).

**Acknowledgments**

We would like to thank the anonymous reviewers for their feedback during the revision process.

This work was supported by the Research Foundation – Flanders (FWO research projects G015020N and 11I7322N).

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

# A    Visualization of confounding mechanism

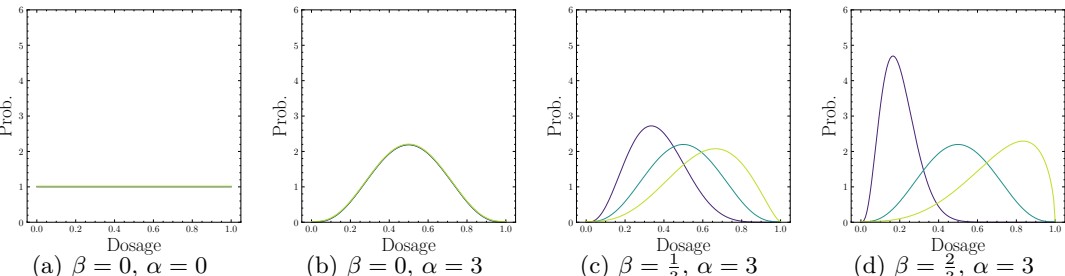

Figure 6: **Visualization of dose distributions per cluster.** Subfigures visualize the dose distributions of different clusters for varying levels of confounding. Parameter $\beta$ determines the difference in doses between individual clusters. Parameter $\alpha$ determines the variability of doses within a cluster.

# B    Effect of regularization on covariate representation

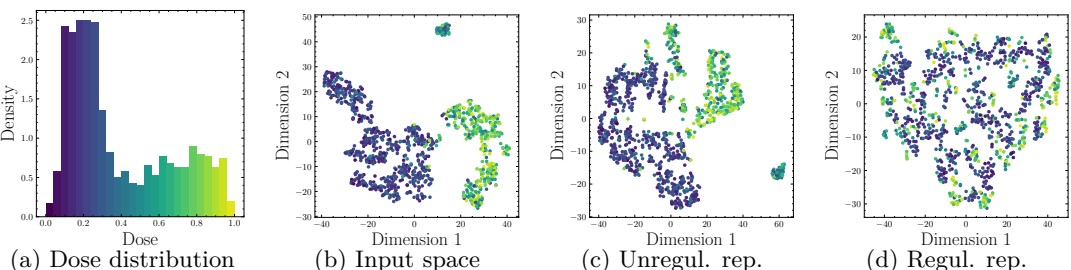

Figure 7: **Regularizing impact of integral probability metric (IPM).** We visualize the regularizing impact of the IPM. Figure (a) visualizes the density of different doses in the test data and provides a color legend. The preceding figures are two-dimensional t-SNE plots of (b) the input space in $\mathcal{X}$, (c) the learned representation of CBRNet without regularization, and (d) the learned representation with regularization by the MMD with a radial basis function kernel. The regularization effectively removes the clustering by dose level.

## C   Extended results

Table 2: **MISE per method on dry bean dataset** for $\beta = 0$ and varying levels of $\alpha$.

$\beta = 0$

| Model | $\alpha$ | | | | |
|---|---|---|---|---|---|
| | 0.0 | 1.0 | 2.0 | 3.0 | 4.0 |
| Linear Regression | $2.83 \pm 0.01$ | $3.27 \pm 0.03$ | $3.80 \pm 0.03$ | $4.19 \pm 0.04$ | $4.44 \pm 0.04$ |
| CART | $0.86 \pm 0.03$ | $1.04 \pm 0.03$ | $1.44 \pm 0.07$ | $1.93 \pm 0.10$ | $2.18 \pm 0.11$ |
| xgboost | $0.55 \pm 0.03$ | $0.69 \pm 0.02$ | $1.11 \pm 0.08$ | $1.58 \pm 0.07$ | $1.88 \pm 0.10$ |
| MLP | $2.83 \pm 0.01$ | $0.54 \pm 0.08$ | $1.17 \pm 0.09$ | $1.98 \pm 0.05$ | $2.52 \pm 0.09$ |
| DRNet | $0.41 \pm 0.03$ | $0.60 \pm 0.06$ | $0.99 \pm 0.12$ | $1.74 \pm 0.19$ | $1.92 \pm 0.23$ |
| VCNet | $0.34 \pm 0.04$ | $0.69 \pm 0.04$ | $1.04 \pm 0.09$ | $1.41 \pm 0.09$ | $1.69 \pm 0.17$ |
| CBRNet($\text{MMD}_{lin}$) | $\mathbf{0.25} \pm 0.04$ | $\mathit{0.35} \pm 0.05$ | $\mathbf{0.41} \pm 0.10$ | $0.64 \pm 0.08$ | $\mathit{0.88} \pm 0.14$ |
| CBRNet($\text{MMD}_{rbf}$) | $\mathit{0.26} \pm 0.04$ | $\mathbf{0.34} \pm 0.05$ | $\mathit{0.42} \pm 0.07$ | $\mathbf{0.61} \pm 0.08$ | $0.92 \pm 0.09$ |
| CBRNet(Wass) | $0.28 \pm 0.05$ | $\mathbf{0.34} \pm 0.05$ | $\mathit{0.42} \pm 0.10$ | $\mathit{0.62} \pm 0.08$ | $\mathbf{0.86} \pm 0.16$ |

Table 3: **MISE per method on dry bean dataset** for $\beta = \frac{1}{4}$ and varying levels of $\alpha$.

$\beta = \frac{1}{4}$

| Model | $\alpha$ | | | |
|---|---|---|---|---|
| | 1.0 | 2.0 | 3.0 | 4.0 |
| Linear Regression | $3.43 \pm 0.03$ | $3.84 \pm 0.04$ | $4.06 \pm 0.03$ | $4.35 \pm 0.04$ |
| CART | $1.17 \pm 0.09$ | $1.47 \pm 0.11$ | $1.74 \pm 0.08$ | $2.04 \pm 0.21$ |
| xgboost | $0.77 \pm 0.07$ | $1.04 \pm 0.07$ | $1.37 \pm 0.09$ | $1.66 \pm 0.13$ |
| MLP | $0.57 \pm 0.06$ | $1.01 \pm 0.11$ | $1.46 \pm 0.13$ | $2.14 \pm 0.13$ |
| DRNet | $0.72 \pm 0.07$ | $\mathit{0.96} \pm 0.07$ | $1.30 \pm 0.16$ | $1.54 \pm 0.20$ |
| VCNet | $0.68 \pm 0.05$ | $1.16 \pm 0.08$ | $1.48 \pm 0.16$ | $1.83 \pm 0.12$ |
| CBRNet($\text{MMD}_{lin}$) | $\mathbf{0.41} \pm 0.09$ | $\mathbf{0.57} \pm 0.17$ | $\mathit{0.71} \pm 0.17$ | $\mathit{0.79} \pm 0.22$ |
| CBRNet($\text{MMD}_{rbf}$) | $\mathit{0.42} \pm 0.11$ | $\mathbf{0.57} \pm 0.15$ | $0.75 \pm 0.26$ | $0.87 \pm 0.15$ |
| CBRNet(Wass) | $0.44 \pm 0.08$ | $\mathbf{0.57} \pm 0.15$ | $\mathbf{0.69} \pm 0.28$ | $\mathbf{0.74} \pm 0.17$ |

Table 4: **MISE per method on dry bean dataset** for $\beta = \frac{3}{4}$ and varying levels of $\alpha$.

$\beta = \frac{3}{4}$

| | $\alpha$ | | | |
|---|---|---|---|---|
| Model | 1.0 | 2.0 | 3.0 | 4.0 |
| Linear Regression | $2.91 \pm 0.01$ | $2.92 \pm 0.02$ | $3.01 \pm 0.03$ | $3.01 \pm 0.02$ |
| CART | $1.24 \pm 0.11$ | $1.31 \pm 0.08$ | $1.32 \pm 0.10$ | $1.43 \pm 0.11$ |
| xgboost | $1.04 \pm 0.05$ | $1.04 \pm 0.08$ | $1.00 \pm 0.06$ | $1.09 \pm 0.06$ |
| MLP | $1.19 \pm 0.09$ | $1.19 \pm 0.1$ | $1.64 \pm 0.11$ | $1.57 \pm 0.15$ |
| DRNet | $0.60 \pm 0.07$ | $0.59 \pm 0.06$ | $0.71 \pm 0.05$ | $0.70 \pm 0.08$ |
| VCNet | $0.55 \pm 0.10$ | $0.62 \pm 0.07$ | $0.60 \pm 0.05$ | $0.66 \pm 0.07$ |
| CBRNet(MMD$_{lin}$) | $0.35 \pm 0.07$ | $0.39 \pm 0.11$ | $\mathbf{0.39} \pm 0.09$ | $\mathbf{0.45} \pm 0.06$ |
| CBRNet(MMD$_{rbf}$) | $0.43 \pm 0.11$ | $0.49 \pm 0.18$ | $0.42 \pm 0.06$ | $0.51 \pm 0.13$ |
| CBRNet(Wass) | $\mathbf{0.34} \pm 0.06$ | $\mathbf{0.38} \pm 0.10$ | $0.42 \pm 0.08$ | $\mathbf{0.45} \pm 0.07$ |

Table 5: **MISE on established benchmark datasets.** CBRNet performs competitively against benchmark methods across established datasets, achieving state-of-the-art performance on three out of four datasets, and improving over a standard MLP on all datasets.

| | Dataset | | | |
|---|---|---|---|---|
| Model | TCGA-2 | IHDP-1 | News-2 | Synth-1 |
| MLP | $2.01 \pm 0.1$ | $2.79 \pm 0.09$ | $1.22 \pm 0.12$ | $1.60 \pm 1.22$ |
| DRNet | $0.35 \pm 0.05$ | $2.54 \pm 0.08$ | $0.81 \pm 0.07$ | $1.52 \pm 1.06$ |
| VCNet | $0.68 \pm 0.30$ | $1.42 \pm 0.19$ | $\mathbf{0.78} \pm 0.06$ | $0.90 \pm 0.55$ |
| CBRNet(MMD$_{lin}$) | $\mathbf{0.24} \pm 0.07$ | $1.60 \pm 0.33$ | $1.12 \pm 0.06$ | $0.99 \pm 0.66$ |
| CBRNet(MMD$_{rbf}$) | $0.26 \pm 0.09$ | $1.56 \pm 0.17$ | $1.10 \pm 0.05$ | $0.99 \pm 0.61$ |
| CBRNet(Wass) | $\mathbf{0.24} \pm 0.08$ | $\mathbf{1.40} \pm 0.25$ | $1.10 \pm 0.05$ | $\mathbf{0.89} \pm 0.56$ |

# D   Dataset ablation

In this section we test the effect of different levels of sparseness in the weight vectors determining the dose response in the Dry bean-DR data (cf. Section 5). For four different levels of sparseness, $\alpha = 3$ and $\beta = \frac{1}{2}$, we generate 10 instances of the Dry bean-DR data. We then train all methods on these instances of the data and report results in Table 6. The results indicate that increasing sparsity does not immediately lead to a decrease in model performance. Across models, performance increases for spareness levels up to 50%. Only for a sparseness of 75% model performance decreases. While CBRNet is similarly affected by high levels of sparsity, the decrease in model performance is less severe when compared with the benchmark methods.

Table 6: **Model performance (MISE) per weight vector sparseness.** We are iterating over different levels of spareness in the weight vectors determining the dose response in the Dry bean-DR data (cf. Section 5). Increasing sparsity does not immediately lead to a decrease in model performance. Across models, performance increases for spareness levels up to 50%. Only for a sparseness of 75% model performance decreases.

$\alpha=3,\ \beta=\frac{1}{2}$

| Sparseness | 0% | 25% | 50% | 75% |
|---|---|---|---|---|
| Linear Regression | $4.88 \pm 0.05$ | $4.39 \pm 0.04$ | $3.78 \pm 0.04$ | $18.91 \pm 0.20$ |
| CART | $2.45 \pm 0.26$ | $1.70 \pm 0.08$ | $1.37 \pm 0.13$ | $11.33 \pm 1.22$ |
| xgboost | $2.25 \pm 0.16$ | $1.55 \pm 0.09$ | $1.23 \pm 0.08$ | $6.25 \pm 0.85$ |
| MLP | $3.09 \pm 0.21$ | $0.98 \pm 0.04$ | $1.31 \pm 0.28$ | $5.41 \pm 0.31$ |
| DRNet | $1.18 \pm 0.05$ | $0.81 \pm 0.06$ | $0.66 \pm 0.06$ | $5.85 \pm 0.30$ |
| VCNet | $1.24 \pm 0.27$ | $0.73 \pm 0.21$ | $0.58 \pm 0.13$ | $5.70 \pm 0.40$ |
| CBRNet(MMD$_{lin}$) | $\mathbf{0.88} \pm 0.13$ | $\mathbf{0.67} \pm 0.12$ | $\mathbf{0.50} \pm 0.29$ | $\mathbf{2.51} \pm 0.82$ |
| CBRNet(MMD$_{rbf}$) | $1.10 \pm 0.30$ | $1.05 \pm 0.28$ | $0.70 \pm 0.35$ | $\mathit{2.71} \pm 1.40$ |
| CBRNet(Wass) | $\mathit{0.90} \pm 0.18$ | $\mathit{0.71} \pm 0.22$ | $\mathit{0.51} \pm 0.25$ | $\mathbf{2.51} \pm 0.87$ |

# E    Implementation, reproducibility, and hyperparameter optimization

Our experiments are written in Python 3.10 ([Van Rossum et al., 1995](#)) and were executed on an Apple M2 Pro SoC with 10 CPU cores, 16 GPU cores, and 16 GB of shared memory. The system needs approximately one day for the iterative execution of all experiments. We aim for maximal reproducibility. All methods considered in our manuscript are consistently implemented in an sklearn style. The code to reproduce all experiments, results, and figures paper can be found online via [https://github.com/christopher-br/CBRNet](https://github.com/christopher-br/CBRNet).

For VCNet, we build on the original implementation provided by [Nie et al. (2021)](#) ([https://github.com/lushleaf/varying-coefficient-net-with-functional-tr](https://github.com/lushleaf/varying-coefficient-net-with-functional-tr)). All remaining neural network architectures were implemented in PyTorch ([Paszke et al., 2017](#)) using Lightning ([Falcon, 2020](#)). Xgboost is implemented using the `xgboost` library ([Chen & Guestrin, 2016](#)). CART models were implemented using the `Scikit-Learn` library ([Pedregosa et al., 2011](#)). Linear regression was implemented using the `statsmodels` library ([Seabold & Perktold, 2010](#)).

For the TCGA-2 dataset, linear regression models were trained using the first 50 principal components of the covariate matrix to reduce computational complexity.

**Hyperparameter optimization.**   Results are not to be compared to the original papers, as the optimization scheme and parameter search ranges differ from the original records. If not specified differently, the remaining hyperparameters are set to match the specifications of the original authors.

Table 7: Hyperparameter search range for Linear Regression:

| Parameter | Values |
| --- | --- |
| Penalty | $\{Elastic\ net, Lasso, None\}$ |

Table 8: Hyperparameter search range for CART:

| Parameter | Values |
| --- | --- |
| Max depth | $\{5, 15, None\}$ |
| Min sample split | $\{2, 5, 20\}$ |
| Min samples per leaf | $\{1, 5, 10\}$ |
| Max features per split | $\{None, \sqrt{p(\mathbf{x})}\}$ |
| Splitting criterion | $\{Gini\}$ |

Table 9: Hyperparameter search range for xgboost:

| Parameter | Values |
| --- | --- |
| Learning rate | $\{0.01, 0.1, 0.2\}$ |
| Max depth | $\{3, 5, 7, 9\}$ |
| Subsample | $\{0.5, 0.7, 1.0\}$ |
| Min child weight | $\{1, 3, 5\}$ |
| Gamma | $\{0.0, 0.1, 0.2\}$ |
| Columns sampled per tree | $\{0.3, 0.5, 0.7\}$ |

Table 10: Hyperparameter search range for MLP:

| Parameter | Values |
|---|---|
| Learning rate | $\{0.0001, 0.001\}$ |
| L2 regularization | $\{0.0, 0.1\}$ |
| Batch size | $\{64, 128\}$ |
| Hidden size | $\{32, 48\}$ |
| Num steps | $\{5000\}$ |
| Num layers | $\{2\}$ |
| Optimizer | $\{Adam\}$ |

Table 11: Hyperparameter search range for DRNet:

| Parameter | Values |
|---|---|
| Learning rate | $\{0.0001, 0.001\}$ |
| L2 regularization | $\{0.0, 0.1\}$ |
| Batch size | $\{64, 128\}$ |
| Hidden size | $\{32, 48\}$ |
| Num dose strata | $\{10\}$ |
| Num steps | $\{5000\}$ |
| Num layers | $\{2\}$ |
| Optimizer | $\{Adam\}$ |

Table 12: Hyperparameter search range for VCNet:

| Parameter | Values |
|---|---|
| Learning rate | $\{0.001, 0.01\}$ |
| Batch size | $\{128, 256\}$ |
| Hidden size | $\{32\}$ |
| Num steps | $\{5000\}$ |
| Optimizer | $\{Adam\}$ |

Table 13: Hyperparameter search range for CBRNet:

| Parameter | Values |
|---|---|
| Learning rate | $\{0.001, 0.01\}$ |
| Batch size | $\{128, 256\}$ |
| Hidden size | $\{32\}$ |
| Num steps | $\{5000\}$ |
| Optimizer | $\{Adam\}$ |
| IPM Regularization | $\{0.001, 0.01, 0.1\}$ |

