# OpenReview forum: "Using representation balancing to learn conditional-average dose responses from clustered data"
_TMLR — Accepted by TMLR_

### Review · Reviewer_vAMe · 2024-08-14

**Summary Of Contributions:**

The paper proposes to use cluster-based ML methods to estimate the conditional average dose response.

**Audience:**

No

**Claims And Evidence:**

No

**Requested Changes:**

The paper is somewhat unclear and requires clarification on several points:

1. The term "conditional-average dose response" is confusing. Typically, "average treatment effect" refers to an expectation taken over certain covariates. In your case, you're conditioning on the covariate, so the natural interpretation would involve taking the expectation over the dosage. However, as defined in Equation (1), this isn't the case—you are estimating the expected response under a fixed dosage and fixed covariate. It might be more accurate to rename this term as "conditional dose response."

2. Assumption 2 states that the covariates X are pretreatment, which is challenging to guarantee in practice. This is a strong assumption and should be explicitly highlighted, in addition to the no hidden confounding assumption, which only implicitly suggests this condition.

3. The paper assumes that "similar units form clusters and that dose assignment is conditional on the cluster." However, in practice, covariate assignment decisions are often based on a subset of covariates. In Section 6, the experiments are shown under a setup where half of the coefficients were set to zero. It would be helpful to vary the sparsity of the weights and report how this affects the performance of the algorithms.

4. The architecture of the estimation procedure in Equation (2) lacks theoretical justification. For instance, you mention that ϕ learns a cluster-agnostic representation, but there is no explanation of which part of the causal quantity this representation is capturing.
Similarly, there is no formal mathematical definition or justification for the statement that "similar units form clusters and the dose assignment is conditional on the cluster." The absence of theoretical guarantees for the proposed estimation procedure weakens the rigor of the paper.

5. It's unclear whether the authors are attempting to learn the number of clusters k in their procedure. If so, the problem is closely related to work on the optimal adjustment set, and baselines from that literature should be included. If not, then the problem loses relevance—practically, one would not know k in advance.

6. The experimental setup is somewhat confusing. It's unclear how S depends on x. Specifically, how does $m_{c_i}$ relate to x as shown in Figure 2?

**Strengths And Weaknesses:**

Overall, the paper would benefit from a more rigorous explanation of the proposed method, along with theoretical guarantees and justifications. Please refer to the detailed requests below.


I may have misunderstood certain aspects of the paper. If the authors could address the requested changes and clarify any misunderstandings on my part, I would be happy to reconsider increasing the score.

---

> ### Author Response · Authors · 2024-10-20
> **Author rebuttal**
>
> Thank you for reviewing our manuscript and for providing feedback, criticisms, and areas for improvement.
> Below, we have taken the time to respond to each of the requested changes.
>
> We hope that these comments and the proposed resolutions address your concern sufficiently.
>
> **Remark 1:** Thank you for the remark.
> The goal of our method is to estimate the response of a certain unit with observed pre-treatment covariates $\mathbf{x}$ to an intervention specified by a dose $s$, which is sometimes referred to as the *”individual”* dose response.
> The term *”individual dose response”*, however, has the potential to be misleading [1], as two units with the same pre-treatment covariates can have different responses to the same dose.
> In this regard, the term *”conditional-average”* is established in the literature [2], though we agree that the term *”average”* can be confusing.
> We would propose to move footnote (1) to the main text and more extensively explain the terminology.
>
> **Remark 2:** Thank you for raising this concern.
> As correctly identified, we expect covariates $\mathbf{X}$ to be pre-treatment and assume strong identifiability, as it is common in much of the related literature, both on estimating dose responses [3], as well as treatment effects [4]. We will highlight this assumption and briefly discuss feasibility.
>
> **Remark 3:** We would like to note that we explain several situations in which intervention assignment is based on clusters in the first paragraph of Section 5.
> Additionally, we would like to clarify that it is not the covariate vector that is partly put to zero, but the coefficients of the dose response function. The inclusion of noise covariates adds to the complexity of CADR estimation [5]. While interesting from a theoretical point of view, we would argue that such an ablation is out of scope for our study.
>
> **Remark 4:** Thank you for the remark.
> We would like to kindly refer to our response to Reviewer AqkL.
>
> **Remark 5:** Thank you for contributing the relation to optimal adjustment sets.
> Indeed, we do not attempt to learn the optimal number of clusters. However, we oppose the view that this weakens the relevance of our method.
> First, our experiments on hyperparameter robustness (Fig. 5b) indicate that increasing $k$ over the number of clusters present in the data does not significantly weaken performance.
> Second, in a real-life scenario, experts might be able to approximate the number of clusters.
>
> A limitation of our approach, however, remains the sample efficiency of the empirical approximations of MMD and Wasserstein distances. We’d gladly add this limitation to the conclusion section of our manuscript.
>
> **Remark 6:** Thank you for the feedback.
> For a unit $i$, $\mathcal{S}i$ is Beta-distributed. To make S dependent on x, we set the mode of the Beta-distribution to $m_i$. $m_i$ is determined by the location of $i$ in the covariate space and we define that every unit in a cluster is assigned the same mode.
>
> We agree with the reviewer that the part of the manuscript is complex, and propose to update the respective passage in the revised version.
>
>
> Once again, thank you for your time, effort, and valuable contribution to our research. We hope that our comments alleviate your concern.
>
> __
> **Sources:**
> [1] Vegetabile. On the distinction between "conditional average treatment effects" (cate) and “individual treatment effects" (ite) under ignorability assumptions. ArXiv. 2021.
> [2] Curth, Svensson, Weatherall and van der Schaar. Really doing great at estimating CATE? a critical look at ML benchmarking practices in treatment effect estimation. NeurIPS. 2021.
> [3] Bica, Jordon, van der Schaar. Estimating the effects of continuous-valued interventions using generative adversarial networks. NeurIPS. 2020.
> [4] Shalit, Johansson, Sontag. Estimating individual treatment effect: generalization bounds and algorithms. NeurIPS. 2017.
> [5] Shi, Blei, and Veitch. Adapting neural networks for the estimation of treatment effects. ArXiv. 2019.

---

> > ### Comment · Reviewer_vAMe · 2024-10-22
> > **updated manuscript request**
> >
> > can the authors provide an updated manuscript, and use a different color to mark the changes made to the original manuscript? Since the paper lacks a theoretical guarantee, a more comprehensive ablation study reflecting the point in remark 3 would be ideal.  Based on the reviews, a clarification of the write-up is needed for the audience to better understand the contribution/limitation of the paper.

---

> ### Comment · Action_Editor_iScJ · 2024-11-13
> **Re: updated manuscript request**
>
> Dear Authors,
>
> Thank you for the discussions.
> > can the authors provide an updated manuscript, and use a different color to mark the changes made to the original manuscript?
>
> To follow up on this comment from Reviewer vAMe, would you like to share an updated version? If so, we can wait until Nov 16.

---

> > ### Author Response · Authors · 2024-11-24
> >
> > Dear editors, dear reviewers,
> >
> > We are happy to take the opportunity to work on an updated version of our manuscript.
> >
> > As discussed with the Action Editor, we will submit this version by the end of December at the latest.
> >
> > Thank you again for your thorough suggestions, critiques and feedback.

---

> > > ### Author Response · Authors · 2024-12-25
> > > **Updated version of the manuscript available**
> > >
> > > Dear Editors and Reviewers,
> > >
> > > We are pleased to inform you that we have uploaded a revised version of our manuscript. To facilitate your review, we have included a version with tracked changes in the supplementary materials.
> > >
> > > We have carefully addressed all comments, feedback, and questions from the previous review round. We look forward to your feedback and remain open to any further suggestions or adjustments needed.
> > >
> > > Thank you for your time and consideration.
> > >
> > > Kind regards,
> > > The Authors

---

> ### Comment · Reviewer_vAMe · 2025-01-03
> **updated paper**
>
> I find the updated paper satisfactory. Recommend accepting.

---

> > ### Author Response · Authors · 2025-01-06
> > **Thank you!**
> >
> > Dear reviewer vAMe,
> >
> > Thank you for checking the revisions in our manuscript and to let us know that our edits were able to clarify your points.
> >
> > If there are any remaining questions, we would be happy to engage further!
> >
> > Best,
> >
> > Authors of Paper3083

---

### Review · Reviewer_AqkL · 2024-08-30

**Summary Of Contributions:**

The authors study estimating conditional average dose response from clustered data where different doses are assigned to different segments of a population.

**Audience:**

Yes

**Claims And Evidence:**

Yes

**Requested Changes:**

Please refer to Weakness.

**Strengths And Weaknesses:**

Strength: On a novel benchmarking dataset, the authors show the impacts of clustered data on model performance and propose an estimator, CBRNet, that learns cluster-agnostic and hence dose-agnostic covariate representations through representation balancing for unbiased CADR inference; they run extensive experiments to illustrate the workings of our method and compare it with the state of the art in ML for CADR estimation.

Weaknesses: NA (updated).

---

> ### Author Response · Authors · 2024-10-20
> **Author rebuttal**
>
> Thank you for the positive review and summary of our contributions. We would like to briefly comment on the inclusion of theoretical results:
>
> The current version of the paper does not focus on the theoretical aspects of the proposed methods, as our key objective is to introduce the method, evaluate its use, and contrast it against state-of-the-art learners. If the reviewer could provide some specific direction on the type of theoretical results (s)he would like us to add, then we would be happy to (try and) provide.
>
> As a side note, we would like to point out that, following the author guidelines and acceptance criteria, TMLR does not require manuscripts to provide theoretical results or guarantees, but *“accurate, convincing and clear evidence”*.
> We believe our experimental analysis provides ample empirical evidence for the viability of our method.

---

### Review · Reviewer_Ytfk · 2024-10-09

**Summary Of Contributions:**

This paper provides a method(CBRNet) estimating CADR from clustered data and where different doses are assigned to different segments of a population. It studies the impacts of learning from clustered data on the performance of ML estimators for CADR and proposes a benchmark to evaluate.

**Audience:**

Yes

**Claims And Evidence:**

Yes

**Requested Changes:**

1. Can the authors explain more about the motivation of using this benchmark and the superiority of it?
2. Can the authors explain how their method is different from other literatures from the causal invariant learning fields?

**Strengths And Weaknesses:**

Strengths:
1 .This paper has clear writing and is easy to follow.

Weakness:
1. The innovation in the paper is not enough in my opinion. Causal invariant learning is studied in many papers. Even though studying it in the case of CADR estimation, it is not by nature very different. The method proposed in this paper is also similar to many used in causal invariant learning.
2. The experiment parts are not convincing enough. It proposed a new benchmarks to evaluate the CADR	estimation. However it is not clear the superiority of this benchmark compared to other common CADR estimation benchmarks.
3. The result table 5 doesn’t show a clear signal of the advantage for CADR estimation method proposed in this paper, as the gap between CADR and the best baseline estimation method is very small.

---

> ### Author Response · Authors · 2024-10-20
> **Author rebuttal**
>
> Thank you for reviewing and providing feedback on our manuscript. Please find our responses below:
>
> **Remark 1:**
> Thank you for the feedback. While causal invariant learning has been extensively studied, our paper offers a novel and meaningful contribution to the field.
> The dose response setting is distinct from the treatment effect setting, as interventions are continuous, not binary.
> While several ML methods exist for tackling causal invariant learning generally, most of those target treatment effect estimation.
> The concepts therein, such as balanced representation learning, are not applicable to CADR estimation, as we lay out in our paper.
>
> Therefore, a key novelty of our method is that by clustering the data, we can apply techniques typically limited to the binary-intervention setting to estimate the effects of continuous-valued interventions.
>
> **Remark 2:**
> Thank you for the remark. The objective for proposing a novel benchmark dataset was by no means to establish a superior benchmark. More so, we believe that there does not and cannot exist a single best benchmark dataset for CADR estimation, or treatment effect estimation in the wider sense.
>
> Our goal is rather to provide a benchmark dataset for types of data-generating processes as described in the paper, so situations in which observations are clustered in the covariate space, and where confounding is driven by clusters. Testing CADR estimators in such scenarios is novel, and has not yet been investigated in the causal ML literature.
>
> **Remark 3:**
> The objective of Table 5 is to provide insight into the general performance of our method, i.e., on datasets that are not necessarily subject to confounding by cluster. Table 5 shows that, generally, CBRNet is competitive on broadly used benchmark datasets (TCGA, News, …). On the other hand, Table 1 provides strong evidence of CBRNet outperforming state-of-the-art methods on a dataset with clustered observations.

---

### Comment · Action_Editor_iScJ · 2025-01-03
**Please check the revised manuscript**

Dear Reviewers,

The authors have provided a revised manuscript:
https://openreview.net/attachment?id=U8EMkndyq4&name=supplementary_material

Could you take a look and see if the changes and the authors' replies address your concerns?

Finally, **please give us updated comments on your recommendations, even if your opinions stay the same**, since we extended the discussion period. (Please make sure your recommendations are in accordance with [Acceptance Criteria of TMLR](https://jmlr.org/tmlr/acceptance-criteria.html).)

Thank you

---

### Comment · Action_Editor_iScJ · 2025-01-07
**A few questions**

Dear Authors,

I have a few questions for clarification.

- I think it would be great if the paper had a formal description of the assumption corresponding to Figure 1 and the following sentences:
"we assume that similar units form clusters and that dose assignment is conditional on the cluster (cf. Figure 1)"
"As we assume that dose assignment is conditional on the cluster"

- Related to the previous point, I'm struggling to understand Figure 1 (c) and the curves of Figure 4. Could the authors add more explanations about the curves, colors, and the dots?

- I'm not convinced by the following claim:
  "As doses are driven by the cluster, traditional supervised estimators, such as feed-forward neural networks might overfit the assignment mechanism and learn biased estimates of dose responses (Schwab et al., 2019)."
  - What are the "traditional supervised estimators" exactly?
  - Why do they tend to overfit when doses are driven by the cluster?
  - What do you mean by "biased" here? Overfitting usually introduces more variance rather than bias.
  - Does (Schwab et al., 2019) claim the same thing? If so, where?

- "we choose one base cluster k": How is it chosen in practice and in the experiments?

- "Different from preceding datasets": which datasets?

---

> ### Author Response · Authors · 2025-01-16
> **Author rebuttal**
>
> Dear editor,
>
> Thank you for these questions and comments, which indicate areas where the paper can benefit from additional detail and clarity. We will address this in a revised version of the paper, as discussed below in our responses to your questions and comments. We will upload this revised version as soon as possible.
>
> __
> ***Remark 1:** I think it would be great if the paper had a formal description of the assumption corresponding to Figure 1 and the following sentences:*
> -  *"we assume that similar units form clusters and that dose assignment is conditional on the cluster (cf. Figure 1)"*
> - *"As we assume that dose assignment is conditional on the cluster"*
>
> **Response:** Such a formal description indeed would provide greater clarity and precision. Therefore, we will add formal descriptions in a revised version of the paper, as we provided for the data generation process in the experiments section. Specifically:
> - Every observation is assigned a cluster $c_i$ by some clustering function $f(x): X \to C$
> - The dose assigned to an observation $i$ is dependent on the cluster of the observation $c_i$, so we can reformulate the no hidden confounders assumption as {$Y(s)|s \in S$}$ ⊥ S|C$
>
> __
> ***Remark 2:** Related to the previous point, I'm struggling to understand Figure 1 (c) and the curves of Figure 4. Could the authors add more explanations about the curves, colors, and the dots?*
>
> **Response:** We will modify the caption of Figure 1 to provide better understanding of Figures 1(c) as follows:
>
> ```
> Figure 1: Illustration of confounding by cluster (CBC).
>
> (a) We assume that there are clusters (green and purple) of similar units (green and purple dots) in the data.
>
> (b) The dose assignment mechanism is a function of the cluster so similar units are assigned similar doses. This results in different distributions of dose assignment per cluster.
>
> (c) As dose responses, y(s), are heterogeneous and depend on a unit’s covariates (x_1, x_2), the dose response curves of units in different clusters (green and purple curves) are different, resulting in confounded data.
> ```
>
> __
> ***Remark 3:** I'm not convinced by the following claim: "As doses are driven by the cluster, traditional supervised estimators, such as feed-forward neural networks might overfit the assignment mechanism and learn biased estimates of dose responses (Schwab et al., 2019)."*
>
> **Response:** We understand and agree that this statement requires further explanation, as we will provide in a revised version of the paper, per the below responses to your specific questions on this statement:
>
> - *What are the "traditional supervised estimators" exactly?*
>
> These are essentially methods, that do not explicitly address the confounding or treatment assignment bias in the data that results from the assignment mechanism in administering doses. For example, the baseline methods used in the experiments (linear regression, CART, XGB, MLP).
>
> - *Why do they tend to overfit when doses are driven by the cluster?*
>
> These methods tend to overfit by memorizing the relation between the dose and the outcome as observed in the available observational data. However, due to the non-random treatment assignment mechanism, some clusters will not be over- or underrepresented at a certain dose, compared to their proportion in the overall population. These traditional supervised methods do not account for confounding, resulting from the dose assignment mechanism (cf. below). Therefore, the resulting models may not generalize well to units of different clusters, with different characteristics and different responses to doses. As illustrated in Figure 1(c), the effect of a particular dose on the outcome is different for observations from a different cluster.
>
> - *What do you mean by "biased" here? Overfitting usually introduces more variance rather than bias.*
>
> We mean assignment bias, resulting from the dose assignment mechanism that results in the observed data. The traditional supervised methods "overfit" to the observed data and do not generalize to the entire population. Here, we use overfitting in the same way as Schwab et al. (2019)—see below.
>
> - *Does (Schwab et al., 2019) claim the same thing? If so, where?*
>
> Schwab et al. effectively discuss on assignment bias and the impact on learning a model from observational data for estimating dose responses, and present a method for addressing assignment bias, i.e., DRNet, as included in the experiments in our paper. From introduction in their paper: "A supervised model naïvely trained to minimise the factual error would overfit to the properties of the treated group, and therefore not generalise to the entire population." We will better position the reference in a revised version of our paper, so as to make clear what exactly is discussed by Schwab in relation to our statement.

---

> > ### Author Response · Authors · 2025-01-18
> > **Updated manuscript available**
> >
> > Dear Editor,
> >
> > Thank you once again for your clarifying questions and constructive remarks.
> > We have just uploaded a revised version of the manuscript.
> >
> > We are looking forward to receiving any additional comments and feedback.
> >
> > Kind regards,
> > The authors

---

> > > ### Comment · Action_Editor_iScJ · 2025-01-21
> > > **"Base cluster"**
> > >
> > > Dear Authors,
> > >
> > > Thank you for the responses and the revision. They have addressed most of my concerns.
> > >
> > > It remains this one:
> > > > "we choose one base cluster k": How is it chosen in practice and in the experiments?
> > >
> > > My understanding is we need to pick one special cluster and name it "cluster k", am I right? I think this is important from a practical point of view and also for reproducibility.

---

> > > > ### Author Response · Authors · 2025-01-21
> > > > **Author rebuttal**
> > > >
> > > > Dear Editor,
> > > >
> > > > Apologies for the confusion. The responses to Remarks 4 and 5 have been cut off due to the character limit.
> > > > Please find out comments below. The updates to the written document will follow asap:
> > > >
> > > > ***Remark 4**: "we choose one base cluster k": How is it chosen in practice and in the experiments?*
> > > >
> > > > **Response**: In practice, one should choose the cluster that includes most observations as the base cluster. As we use empirical approximations of the IPMs, this ensures good approximation. We will indicate and motivate this choice in the revised version of the paper.
> > > >
> > > > __
> > > > ***Remark 5**: "Different from preceding datasets": which datasets?*
> > > >
> > > > **Response:** Here, we mean the datasets that have been used in previous papers on CADR estimation. These datasets are used as well in the benchmarking experiment presented in Sections 5-6 of our paper. We will indicate this in a revised version of the paper and add references.

---

> > > > > ### Comment · Action_Editor_iScJ · 2025-01-25
> > > > > **Small question**
> > > > >
> > > > > Thank you for your response.
> > > > >
> > > > > Another question: is this citation correct?
> > > > > > we take inspiration from Schwab et al. (2018) (before Eq. 4).

---

> > > > > > ### Author Response · Authors · 2025-01-26
> > > > > > **Author rebuttal**
> > > > > >
> > > > > > Dear Editor,
> > > > > >
> > > > > > Thank you for your inquiry.
> > > > > >
> > > > > > We drew inspiration from Schwab et al.'s extension of the PEHE metric (Precision in Estimating Heterogeneous Effects) to multiple treatments, as described at the end of Section 3 in their manuscript.
> > > > > >
> > > > > > Since PEHE is defined only for binary treatments, the authors proposed averaging the PEHE values across all possible pairs of treatments.
> > > > > >
> > > > > > We adopted a similar approach to address the limitation of IPMs, which are defined only for two distributions. This methodology led to Eq. (4) in our manuscript. As such, the citation is accurate.
> > > > > >
> > > > > > Please let us know if you require further details or clarification.

---

### Decision · Action_Editor_iScJ · 2025-01-25

**Recommendation:** Accept with minor revision

**Comment:**

The recommendations from the three reviewers were one vote for "leaning accept" and two for "leaning reject".

However, this submission went through a little unusual process of review because of the extension of the discussion period after the reviewers posted recommendations. After the discussions, most of the concerns seem to have addressed, and I recommend acceptance with minor revisions, according to the exchanges between the authors and the reviews including myself.

There is a few very small concerns about a citation and an equation format, which should be easily addressed in preparing the final version.

**Audience:**

The tackled problem and the proposed method are highly relevant in many practical applications that should interest researchers of TMLR's audience. The authors provide examples and their importance, for example, in paragraph Motivation of Section 4, by citing several previous studies.

**Claims And Evidence:**

This paper works on the task of estimating the conditional-average dose response, which is the conditional-average treatment effect of a continuous treatment variable.
  The authors propose a method for this task under the assumption that the data points form clusters, and these clusters are the only confounders for the potential outcome and the treatment assignment. This can be seen as a modified condition of the standard assumption in Assumption 2 (No hidden confounders) described in the paper.
  The proposed method trains a neural network by minimizing the mean squared error of its predictions of the responses while minimizing the discrepancy between the distributions of the extracted features for different clusters. The latter part of the loss encourages the representations used in the predictions to be invariant to the differences of clusters so that the model will not rely on the confounders in prediction, as was done by Johansson et al. (2016) and Shalit et al. (2016).

  The authors test the proposed method with their novel semi-synthetic benchmark data and previously used datasets and confirm its effectiveness. Some extended results including ablation studies can be found in the appendices.

---

> ### Author Response · Authors · 2025-02-03
> **Camera-ready manuscript available**
>
> Dear Editor and Reviewers,
>
> We have uploaded the camera-ready version of our manuscript. In response to your decision letter, we have clarified in Section 4 how our methodology, particularly Equation (4), was motivated by Schwab et al. (2018).
>
> Once again, we sincerely appreciate your time, effort, and valuable feedback throughout the review and submission process. We are confident that your input has significantly improved our work.
>
> Kind regards,
> The Authors